# OpenReview forum: "metabench - A Sparse Benchmark of Reasoning and Knowledge in Large Language Models"
_ICLR.cc/2025/Conference — ICLR 2025 Poster_

### Official Review · Reviewer_Fxth · 2024-11-04

**Soundness:** 2
**Presentation:** 2
**Contribution:** 2
**Rating:** 5
**Confidence:** 2

**Summary:**

This paper proposes metabench, a compressed version of six popular LLM benchmarks (ARC, GSM8K, HellaSwag, MMLU, TruthfulQA, and WinoGrande) that achieves comparable evaluation capability while using less than 3% of the original items. The authors leverage psychometric techniques, particularly Item Response Theory (IRT), to identify the most informative test items and estimate latent abilities that can reconstruct original benchmark scores with high accuracy.

**Strengths:**

1. Novel application of psychometric methods to LLM evaluation
2. Impressive compression ratio (<3% of original size) while maintaining accuracy. Low reconstruction error (1.24% RMSE for individual benchmarks, 0.58% for total score).
3. Comprehensive ablation studies and baseline comparisons. Thorough investigation of factor structure across benchmarks.

**Weaknesses:**

1. Memorization risks:
    (1) Smaller benchmark size increases memorization vulnerability
    (2) Proposed mitigation strategies need further validation
2. Theoretical Assumptions:
    (1) IRT assumptions about LLMs need more justification
    (2) Independence assumptions between models may be violated due to shared architectures/training data

**Questions:**

1. Could alternative item selection methods (beyond Fisher information) yield better results?
2. How stable are the results across different random seeds and model subsets?

---

> ### Author Response · Authors · 2024-11-21
>
> [Part 1/2]
>
> We thank the reviewer for their detailed and motivating review of our paper. We are glad the reviewer found our results impressive and our analyses comprehensive and thorough. The reviewer also provided crucial pointers for further analyses to prove the soundness of our methodology. In particular, we tested (1) the validity of our memorization mitigation strategies, (2) the performance of an alternative item selection procedure, and (3) the robustness of our results with regard to random seeds.
>
> > Memorization risks: (1) Smaller benchmark size increases memorization vulnerability (2) Proposed mitigation strategies need further validation
>
> The reviewer is correct to remark on the problem of memorization and contamination, and we have now performed some experiments to validate our mitigation strategies. We have included a discussion of this in a new section (Section 3. Using metabench), complementing the discussion in section B.6 of the appendix. Specifically, we have created four versions of metabench: two disjoint sets of distinct items from each benchmark (validation version vs. test version), and versions of these where choices in multiple-choice questions are fixed or are randomly re-labelled (standard choices vs. permuted choices).
>
> In section 3, we present an experiment testing the effect of memorization on performance on each of these sets. We evaluated three LLMs on each version of metabench, then fine-tuned them on the validation set of items with fixed multiple choice orders. We then evaluated them on all four versions again. While performance on the finetuning set significantly improved, this performance boost was attenuated by permuting the answer choices, and performance did not improve significantly on the disjoint test set.
>
> This suggests that having four versions of metabench can go some way to mitigating the problems of memorization and contamination, although of course we must ultimately trust that researchers would never explicitly train on any component of metabench if they want to use it to make meaningful inferences about performance or ability.
>
>
> > Theoretical Assumptions: (1) IRT assumptions about LLMs need more justification (2) Independence assumptions between models may be violated due to shared architectures/training data
>
> We refer the reviewer to section B.4 of the appendix for a thorough justification of the use of IRT on artificial agents. In short, item response theory is a statistical method to derive properties of test items based on observed response patterns. It makes no assumptions about the nature of the agents tested using the items [1], and many authors in the ML community have adopted this method already [2-7].
>
> IRT does not assume marginally independent responses across agents, but conditionally independent ones (conditioned on the latent ability). In appendix B.8, we show that the reconstruction performance of metabench is unbiased wrt. to LLM architecture. Furthermore, one cannot test for overlap in training data across models. However, analogously, students often have largely overlapping “training data” due to pre-specified educational curricula - and still IRT is used successfully on student populations. In summary, this suggests that if architecture or training data create substantial dependence in response accuracies not captured by a benchmark’s corresponding latent ability, the performance of metabench does not suffer from it.

---

> ### Author Response · Authors · 2024-11-21
>
> [Part 2/2]
>
> > Could alternative item selection methods (beyond Fisher information) yield better results?
>
> Thank you for suggesting this additional experiment consolidating our methodological choices. We will add the following contents to the appendix. The corresponding code is contained in “clustering.R” and “evaluate.clust.R” in the Supplementary Material.
>
> An alternative item selection strategy presented in previous literature (e.g. [7]) is based on clustering. We altered only the item selection method and kept the remaining analysis steps constant for maximum comparability with our previous results: For this, we took the IRT-based item parameters from the 350-item fits for each benchmark and performed k-means clustering on them. Iteratively, one item was drawn from each remaining cluster, until the total number of items equaled the number of items in our Fisher-information-based subsets. Then we re-fitted the same IRT model on that subset of items, estimated the latent ability per LLM and fitted a GAM of the original score based on the latent ability. We performed a grid search over the hyperparameters k in [10, 15, 20, 25, 30], IRT model type m in [2PL, 3PL, 4PL] and latent ability estimation method tau in [MAP, EAPsum] on a validation set. For comparability with our final selection, we then took the item selection with the best validation RMSE for each benchmark and fit a joint GAM (cf. appendix B.5) on the same test set as used for our main results. These are the test RMSEs:
>
> | _Benchmark_   | ARC     | GSM8K   | HellaSwag | MMLU   | TruthfulQA | WinoGrande | Total Score |
> |---------|---------|---------|-----------|--------|------------|------------|-------------|
> | _Test RMSE_| 1.334   | 1.748   | 1.697     | 1.805  | 1.293      | 2.494      | 0.775       |
>
> Each test RMSE is substantially worse than its counterpart with item selection based on Fisher-information (s. Paper Figure 1). Note that since we kept the remaining variables in the processing pipeline constant, this highlights the merit of using Fisher-information functions for item selection.
>
> > How stable are the results across different random seeds and model subsets?
>
> We acknowledge our oversight in not conducting this analysis sooner and thank the reviewer for the pointer. The following results will get a separate section in the appendix. We ran the entire benchmark distillation procedure using 5 different random seeds, which affect the dataset partitioning into training, validation and test sets, as well as cross-validated subsampling. Please find the results in rebuttals Figure 1 in the Supplementary Material. Test set RMSEs and MAEs show little variation, but subsets for ARC and MMLU are slightly less stable than for the other benchmarks or the total score over benchmarks. Rank correlations are always over r = 0.95, and only MMLU and WinoGrande do not border on r = 1.0. Apart from two outliers of ~50 items, the converged benchmark sizes are largely stable as well. Overall, the results seem largely independent of the chosen random seed.
>
> ---
>
> __References__
>
> [1] Paul Irwing, Tom Booth, and David J. Hughes (eds.). The Wiley Handbook of Psychometric Testing: A Multidisciplinary Reference on Survey, Scale and Test Development. Wiley, 1 edition, April
> 2018. ISBN 978-1-118-48983-3 978-1-118-48977-2. doi: 10.1002/9781118489772
>
> [2] Fernando Martínez-Plumed, Ricardo BC Prudêncio, Adolfo Martínez-Usó, and José Hernández-
> Orallo. Making sense of item response theory in machine learning. In ECAI 2016, pp. 1140–1148.
> IOS Press, 2016.
>
> [3] Fernando Martínez-Plumed, Ricardo B.C. Prudêncio, Adolfo Martínez-Usó, and José Hernández-
> Orallo. Item response theory in AI: Analysing machine learning classifiers at the instance level.
> Artificial Intelligence, 271:18–42, June 2019. ISSN 00043702. doi: 10.1016/j.artint.2018.09.004.
>
> [4] Clara Vania, Phu Mon Htut, William Huang, Dhara Mungra, Richard Yuanzhe Pang, Jason Phang,
> Haokun Liu, Kyunghyun Cho, and Samuel R. Bowman. Comparing Test Sets with Item Response
> Theory, June 2021.
>
> [5] Xiting Wang, Liming Jiang, Jose Hernandez-Orallo, David Stillwell, Luning Sun, Fang Luo, and
> Xing Xie. Evaluating General-Purpose AI with Psychometrics, December 2023a.
>
> [6] Hernández-Orallo, J. (2017). Evaluation in artificial intelligence: from task-oriented to ability-oriented measurement. Artificial Intelligence Review, 48, 397-447.
>
> [7] Felipe Maia Polo, Lucas Weber, Leshem Choshen, Yuekai Sun, Gongjun Xu, and Mikhail Yurochkin. tinyBenchmarks: Evaluating LLMs with fewer examples, February 2024.

---

### Official Review · Reviewer_cph2 · 2024-11-04

**Soundness:** 4
**Presentation:** 3
**Contribution:** 4
**Rating:** 6
**Confidence:** 3

**Summary:**

The paper introduces Metabench, a sparse benchmarking method designed to evaluate large language models (LLMs) with minimal redundancy and resource demands. By analyzing data from over 5000 LLMs across six benchmarks (ARC, GSM8K, HellaSwag, MMLU, TruthfulQA, and WinoGrande), Metabench distills these into a much smaller subset, reducing the combined item count by over 97%. Using psychometric techniques such as Item Response Theory (IRT), Metabench selects the most informative items, facilitating efficient and accurate evaluation while maintaining the integrity of the original benchmarks. The sparse benchmark achieves impressive fidelity, reconstructing original scores with less than 1.24% RMSE on average, and identifying a single common latent factor strongly correlating with general model ability.

**Strengths:**

1. The paper’s technical approach is methodologically sound, with robust use of IRT and statistical modeling to identify informative items.
2. It is well-organized, with a clear explanation of Metabench’s goals and psychometric techniques.
3. It makes a substantial contribution to LLM evaluation, providing a novel, efficient, and scalable benchmarking solution.

**Weaknesses:**

1. The framework currently focuses on six benchmarks; additional work could explore its applicability across a broader range of LLM tasks or domains.
2. Metabench’s dependence on psychometric models, especially IRT, could be limiting if these models do not fully capture the complexities of LLM behavior, as they were traditionally designed for human subjects.

**Questions:**

1. Could authors elaborate on potential limitations when applying Metabench to other domains?
2. How might Metabench handle scenarios where specific benchmarks assess unique skills not captured by a general latent factor?

---

> ### Author Response · Authors · 2024-11-21
>
> We thank the reviewer for their rewarding and thorough review of our paper. We are happy the reviewer found our methods sound, our paper well-organized, and acknowledged our contribution as substantial. The reviewer also inspired an additional simulation experiment to test under which circumstances our score reconstruction approach can handle latent abilities that are not well covered by a general latent factor.
> > The framework currently focuses on six benchmarks; additional work could explore its applicability across a broader range of LLM tasks or domains. (...) Could authors elaborate on potential limitations when applying Metabench to other domains?
>
> We hope that the methods and ideas used to create metabench will act as stepping stones for future research on resource-efficient AI benchmarking. There are three important conditions under which applying our methods to a different set of benchmarks promises to be successful:
> 1. IRT-based benchmark distillation requires the availability of large datasets containing single-item accuracies for benchmarks run by thousands of LLMs.
> 1. A reducible benchmark needs to be large and specific enough, such that there is enough information overlap between items to create exploitable redundancy.
> 1. For synergy effects, a substantial number of LLMs needs to be evaluated on multiple benchmarks, such that the correlation structure among benchmark scores can be used for joint score recovery.
>
> However, since IRT is only based on the accuracy of multiple choice questions, in principle any classification benchmark with quantifiable accuracy can be reduced under the conditions named above. For instance, as an example for large computer-vision datasets, [1] apply IRT on the ImageNet validation set. While they show that IRT methods can be used to distil the validation set into informative subsets, the potential to reduce the entire benchmark remains unexplored to the best of our knowledge (see also [2] for an overview on pruning methods for ImageNet).
>
>
> > Metabench’s dependence on psychometric models, especially IRT, could be limiting if these models do not fully capture the complexities of LLM behavior, as they were traditionally designed for human subjects.
>
> We refer the reviewer to section B.4 of the appendix for a thorough justification of the use of IRT on artificial agents. In short, item response theory is a statistical method to derive properties of test items based on observed response patterns. In that sense, IRT models only aim to provide a summary of test behavior rather than capture some complex qualitative patterns. Furthermore, IRT makes no assumptions about the nature of the agents tested using the items [3], and many authors in the ML community have adopted this method already as a more nuanced way to analyze benchmark performance [4-9].
>
>
> > How might Metabench handle scenarios where specific benchmarks assess unique skills not captured by a general latent factor?
>
> This is a great question and we conducted a proof of concept simulation study to address it: We repeatedly simulated two latent abilities for 500 subjects and two benchmarks of 100 items. The first benchmark measures both abilities to a varying degree, and the second test only measures the second ability. We separately varied the correlation between both latent abilities and the extent to which the first benchmark depends on both latent abilities (more details are contained under Rebuttals Figure 2 in the Supplementary Material). If two latent abilities are weakly correlated, they cannot be well described by a general latent factor. Our simulation shows that in this case, score reconstruction for the first benchmark benefits from including the latent ability estimated from the second benchmark, especially if the first benchmark moderately measures the second ability. This suggests that using multiple latent abilities (each derived from a different benchmark) to reconstruct the score of a single benchmark is the way to go in this scenario. Thus, metabench can indeed handle scenarios where specific benchmarks assess unique skills not captured by a general latent factor.

---

> ### Author Response · Authors · 2024-11-21
>
> **References**
>
> [1] Ramachandran, R., Kulkarni, T., Sharma, C., Vijaykeerthy, D., & Balasubramanian, V. N. (2024). On Evaluation of Vision Datasets and Models using Human Competency Frameworks. arXiv preprint arXiv:2409.04041.
>
> [2] Sorscher, B., Geirhos, R., Shekhar, S., Ganguli, S., & Morcos, A. (2022). Beyond neural scaling laws: beating power law scaling via data pruning. Advances in Neural Information Processing Systems, 35, 19523-19536.
>
> [3] Paul Irwing, Tom Booth, and David J. Hughes (eds.). The Wiley Handbook of Psychometric Testing: A Multidisciplinary Reference on Survey, Scale and Test Development. Wiley, 1 edition, April
> 2018. ISBN 978-1-118-48983-3 978-1-118-48977-2. doi: 10.1002/9781118489772
>
> [4] Fernando Martínez-Plumed, Ricardo BC Prudêncio, Adolfo Martínez-Usó, and José Hernández-
> Orallo. Making sense of item response theory in machine learning. In ECAI 2016, pp. 1140–1148.
> IOS Press, 2016.
>
> [5] Fernando Martínez-Plumed, Ricardo B.C. Prudêncio, Adolfo Martínez-Usó, and José Hernández-
> Orallo. Item response theory in AI: Analysing machine learning classifiers at the instance level.
> Artificial Intelligence, 271:18–42, June 2019. ISSN 00043702. doi: 10.1016/j.artint.2018.09.004.
>
> [6] Pedro Rodriguez, Joe Barrow, Alexander Miserlis Hoyle, John P. Lalor, Robin Jia, and Jordan
> Boyd-Graber. Evaluation Examples are not Equally Informative: How should that change NLP
> Leaderboards? In Proceedings of the 59th Annual Meeting of the Association for Computational
> Linguistics and the 11th International Joint Conference on Natural Language Processing (Volume
> 1: Long Papers), pp. 4486–4503, Online, 2021. Association for Computational Linguistics. doi:
> 10.18653/v1/2021.acl-long.346.
>
> [7] Clara Vania, Phu Mon Htut, William Huang, Dhara Mungra, Richard Yuanzhe Pang, Jason Phang,
> Haokun Liu, Kyunghyun Cho, and Samuel R. Bowman. Comparing Test Sets with Item Response
> Theory, June 2021.
>
> [8] Xiting Wang, Liming Jiang, Jose Hernandez-Orallo, David Stillwell, Luning Sun, Fang Luo, and
> Xing Xie. Evaluating General-Purpose AI with Psychometrics, December 2023a.
>
> [9] Hernández-Orallo, J. (2017). Evaluation in artificial intelligence: from task-oriented to ability-oriented measurement. Artificial Intelligence Review, 48, 397-447.

---

> > ### Comment · Reviewer_cph2 · 2024-11-30
> >
> > Thanks for the detailed response. I'll keep my score as is.

---

### Official Review · Reviewer_iUxe · 2024-11-04

**Soundness:** 3
**Presentation:** 3
**Contribution:** 3
**Rating:** 6
**Confidence:** 2

**Summary:**

This paper introduces metabench, a sparse benchmark distilled from six prominent benchmarks (ARC, GSM8K, HellaSwag, MMLU, TruthfulQA, and WinoGrande). Simple criteria, cross-validated subsampling, and information-based filtering are used to reduce the size of the benchmark. Original scores are reconstructed in a cross-validated manner.

**Strengths:**

1. This paper distills six prominent LLM benchmarks into a much smaller one with less than 3% of the size, which enables more streamlined and cost-effective evaluation methods;
2. The new sparse benchmark yields estimators able to reconstruct the original benchmark score.

**Weaknesses:**

As mentioned in the limitations section, a smaller benchmark has the risk of being memorized.

**Questions:**

Will a small benchmark lead to a large variance in evaluation?

---

> ### Author Response · Authors · 2024-11-21
>
> We are happy about the positive review of our paper and thank the reviewer for motivating us to further investigate whether our mitigation strategies against memorization are valid.
>
> > As mentioned in the limitations section, a smaller benchmark has the risk of being memorized.
>
> The reviewer is correct to remark on the problem of memorization and contamination. We have included a discussion of this in a new section (3. Using metabench), complementing the discussion in the section B.6 of the appendix. Specifically, we have created four versions of metabench:
> - Version A: The main set of items from each benchmark as presented in the paper.
> - Version B: Another similarly sized set of items from each benchmark with 0 overlap with version A (useful when an LLM is evaluated multiple times on version A)
> - Versions A and B, in which the answer choices are randomly permuted (thus, if an LLM has memorized only the response number, it should fail to choose the new correct option)
>
> In section 3, we present an experiment testing the effect of memorization on performance on each of these sets. The analysis consists of three stages.
> 1. We evaluated three LLMs on each version of metabench.
> 1. We fine-tuned the on version A with fixed multiple choice orders.
> 1. We then evaluated them on all four versions again.
>
> While performance on the training set significantly improved, as expected, this performance boost was attenuated by permuting the answer choices, and performance did not improve significantly on the disjoint set. Therefore, having four versions of metabench can go some way to mitigating the problems of memorization and contamination, although of course we must ultimately trust that researchers would never explicitly train on any component of metabench if they want to use it to make meaningful inferences about performance or ability.
>
>
>
> > Will a small benchmark lead to a large variance in evaluation?
>
> This is a crucial point and we thank the reviewer for drawing our attention to it. If a regression model is unbiased (its average residual is 0), then its MSE is identical with the prediction error variance. Thus, the RMSE is identical with the standard deviation of the error variance. In the paper we show how the test RMSE grows _sublinearly_ with the size of cv-subsampled benchmarks (Section 2.2 and Appendix B.2): While the error variance increases with shrinking benchmark sizes, it does so very slowly until a certain point. Because there is so much overlap in the information measured by all items, we lose very little by discarding a large proportion of benchmark items. The important aspect is that the remaining items cover enough space of the ability landscape tested by the benchmark. This is exactly why we use information filtering to construct metabench.

---

> > ### Comment · Reviewer_iUxe · 2024-12-02
> >
> > Thank you for your clarification. I will keep my score.

---

### Official Review · Reviewer_rtaB · 2024-11-05

**Soundness:** 4
**Presentation:** 4
**Contribution:** 3
**Rating:** 8
**Confidence:** 3

**Summary:**

This paper considers the six LLM benchmarks included in the Open LLM Leaderboard (ARC, GSM8K, HellaSwag, MMLU, TruthfulQA, and WinoGrande) and seeks to create a much smaller benchmark that is predictive of the original suite by subselecting items. This is done using data from more than 5000 LLMs included in the leaderboard and a pyschometric method called item response theory (IRT) which in essence fits a model that estimates the item's difficulty and how well the item discrimates between models whose "abilities" are close to the item's difficulty. (Note this model ability is also fit by the method in an alternating fashion.) The presented method results in a benchmark that is only 3% the size of the original benchmark but is able to effectively reconstruct both the original individual benchmark scores and the joint score. Finally, using factor analysis, the authors demonstate that a single latent is predictive of all 6 benchmarks.

**Strengths:**

The paper is well-written and clearly communicates its ideas and methods. In the choice of the IRT, multiple models and methods for estimating the ability are explored. The method proposed produces a much smaller benchmark which the authors demonstrate has better predictive power than randomly subsampling items (Figure 1B). Careful consideration is given to potential limitations of the method, including assumptions about the conditional independence of the LLMs used for the study. The work also considers the interesting idea of a benchmark that performs adaptive testing in which items are selected sequentially based on a current estimate of the model's ability.

Overall I think the paper makes meaningful contributions to studying LLM benchmarks and making model evaluation more efficient, and I thus lean towards acceptance. However, I do think the benchmarks considered are missing some of the abilities that people seek to measure in LLMs (e.g. coding), somewhat limiting the work's impact. I seek to provide concrete suggestions regarding this in the next section.

**Weaknesses:**

My comments in this section are not intended to be required changes to the paper but rather a discussion of what I think the authors could add to have more significant impact.

Currently the main output of the paper is a much smaller benchmark that can be used to efficiently rank models on the six benchmarks as well as evidence from factor analysis that all six benchmarks are measuring a single latent ability. However, across the broader field of LLM benchmarks, it is generally assumed that there are multiple latent dimensions to the abilities of LLMs. For example, if a code benchmark was added into the set, I would assume this would require another latent dimension to fit model performance, and it would be intriguing if this was not true! Also I would be curious if a larger fraction of the test items is required to reconstruct the scores when the set of included benchmarks require multiple latent ability dimensions to represent.

In essence, the most interesting direction I see for this work is to apply the methods to a more comprehensive set of benchmarks to try to discover latent ability dimensions that might be interpretable as what we think of as LLM capabilities. This should then also provide a characterization of which of these abilities each benchmark measures.

**Questions:**

For quite a few models on the leaderboard, the MMLU score will be random chance (~25%, which you can see in Figure 1). Would it be a useful preprocessing step to subtract out random chance from the score and renormalize? E.g. take (score - 0.25) / (1 - 0.25).

---

> ### Author Response · Authors · 2024-11-21
>
> [Part 1/2]
>
> We thank the reviewer for their thoughtful and encouraging review of our paper. We are glad the reviewer found our paper well-written and carefully thought through. The reviewer also inspired further investigation on the dimensionality and broadness of latent abilities.
> > if a code benchmark was added into the set, I would assume this would require another latent dimension to fit model performance
>
> Thank you for raising this interesting question, it is very plausible that performance on a coding benchmark does not only depend on some general ability to solve analytical problems, but that it would also depend on knowledge in programming languages, program architecture, data structures etc. In that case, a 2-dimensional IRT-model (2 latent abilities) is likely superior to a one-dimensional IRT model to reconstruct a coding benchmark’s score. However, It is unclear how much unique structure this would add to the covariance matrix across latent abilities for multiple benchmarks - that is, if that would warrant a second latent ability in FA.
>
> In fact, your question points towards a broader topic. There’s an important distinction to make between our IRT-based benchmark distillation and the FA results:
> IRT: For each benchmark, we essentially fit a distinct one-dimensional IRT model aiming to capture the latent ability tested by the corresponding benchmark. One can view it as the test-specific aptitude. This approach yields six distinct latent abilities.
> FA: We show with FA that these abilities are largely governed by one more abstract ability. However, we do not enforce this dependence structure through our analysis choices: We do not fit a single IRT model on all benchmarks at once to find one-dimensional latent ability that is captured jointly by all six benchmarks.
>
> This, on the other hand, raises the question of why we fit one-dimensional IRT models. We added the following paragraph to the appendix and add a reference to it in Section 2.3:
>
> _How many distinct abilities play into solving a benchmark?_ This is a nuanced question and is distinct from _how many ability dimensions are sufficient for the purpose of score reconstruction_. While conceptually it is a hard claim that a single ability governs test performance, there are three reasons for using one-dimensional IRT models in our pipeline:
> 1. _Item selection_: Information filtering scales well in the one-dimensional case. Finding the most-informative items in a multi-dimensional space opens up new problems: Do we marginalize out each dimension? Do we search in the joint space? If yes, what information coverage is desirable, which parts of the n-dimensional grid are most relevant overall?
> 1. _Estimation variance_: The total number of loading parameters doubles with each added latent ability, which increases the uncertainty in the IRT fit, on which we base the remaining selection process.
> 1. _Diminishing returns_:  Table 5 shows that for score reconstruction, the performance boost from using two latent abilities is negligible (when present at all).
>
> Table 5: _Two-dimensional latent abilities do not substantially aid score recovery_. For the 350-item version of each benchmark, we fit a 1-dimensional and a 2-dimensional 2PL model and derived MAP estimates of the latent abilities. We then fit a GAM of the original score using either the single latent ability from the 1-dim fit or both latent abilities from the 2-dimensional fit. RMSEs are reported on identical test sets per benchmark.
>
> | | ARC | GSM8K | HellaSwag | MMLU | TruthfulQA | WinoGrande |
> |---|---|---|---|---|---|---|
> | RMSE(1-dim) | 0.893 | 1.293 | 0.844 | 1.048 | 0.988 | 1.055 |
> | RMSE(2-dim) | 0.893 | 1.253 | 0.820 | 1.065 | 0.989 | 1.003 |
> | Δ | 0.000 | 0.040 | 0.024 | -0.017 | -0.001 | 0.052 |

---

> ### Author Response · Authors · 2024-11-21
>
> [Part 2/2]
>
> > Would it be a useful preprocessing step to subtract out random chance from the score and renormalize? E.g. take (score - 0.25) / (1 - 0.25).
>
> Thank you for this suggestion. We have given this careful thought and came to the following conclusions: It is better not to normalize for guess-rate to keep normalized scores comparable across studies. In a few LLMs this would in fact lead to negative normalized scores, especially in GSM8K.
> As score normalization does not alter the single responses, IRT fits would remain unaffected. Finally, since this normalization procedure is a fixed affine transform, it can be perfectly adapted to by the weights of a regression model (like the GAMs we use). It would therefore not affect score reconstruction performance either.
>
> > Also I would be curious if a larger fraction of the test items is required to reconstruct the scores when the set of included benchmarks require multiple latent ability dimensions to represent.
>
> This is a nuanced question: In order to answer this, we conducted a proof of concept simulation study, in which we repeatedly simulated two latent abilities for 500 subjects and two benchmarks of 100 items. The first benchmark measures both abilities to a varying degree, and the second test only measures the second ability. We separately varied the correlation between both latent abilities and the extent to which the first benchmark depends on both latent abilities (more details are contained under Rebuttals Figure 2 in the Supplementary Material). In summary, if a test requires multiple latent abilities and they are not strongly correlated, score recovery benefits from adding multiple latent abilities as regression predictors. Therefore (assuming independent loading parameters over dimensions) one needs enough items to estimate each of the required latent abilities, overall raising the number of required items. So your hunch is probably correct!

---

> > ### Comment · Reviewer_rtaB · 2024-11-26
> > **Response to Authors**
> >
> > I thank the authors for their response and the substantial revisions to the paper. I especially appreciate two of the major additions which (1) test whether any improvement comes from adding a second latent ability for the set of 6 benchmarks considering in the paper and (2) explore through simulation the effects of having two latent abilities and two benchmarks with varying relations to the latent abilities. While I remain interested in seeing an expanded family of benchmarks for which a second latent does provide an improvement, I view this as a substantial expansion to the work and thus beyond the scope of the paper. Based on the improvements made by the authors, I have raised my score.

---

> > > ### Author Response · Authors · 2024-11-26
> > >
> > > Thank you very much for your constructive review, your fruitful suggestions, and for raising the score to an 8. We deeply appreciate it!

---

### Author Response · Authors · 2024-11-21

We thank all reviewers for their constructive and helpful feedback. Their input was immensely valuable to further improve our submission. The reviewers’ assessment was overall positive with only one reviewer giving a score of 5:
- Reviewer iUxe deemed our methods as “streamlined and cost-effective”.
- Reviewer Fxth found our analyses “comprehensive [and] thorough” and deemed our results “impressive”.
- Reviewer rtaB praised our paper as “well-written [and] clear". They further assessed that our paper and benchmark are "a meaningful contribution[...] to studying LLM benchmarks and making model evaluation more efficient”.
- Reviewer cph2 added that our paper is “well-organized, with a clear explanation of goals and [...] techniques”. They also acknowledged our contribution as “substantial [for the field of] LLM evaluation”.

In response to the reviewers' feedback, we have made the following major additions:
1. We re-ran the entire pipeline for benchmark distillation with 4 additional seeds and checked the stability of our reported results with regard to test error, rank preservation and benchmark sizes.
2. We implemented a clustering-based item selection technique whose performance we compared with our information filtering approach, while keeping all other variables constant.
3. We fit 2-dimensional IRT models and tested if adding a second latent ability significantly improved score reconstruction.
4. We validated our strategies against test memorization by fine-tuning 3 LLMs on the main set of metabench and testing its performance on the disjoint alternative set and on both sets with permuted answer labels.
5. We ran a simulation study that reveals, under which conditions synergy effects across benchmarks arise for score reconstruction.

These analyses, its results, and answers to further reviewer questions are contained in our responses to the individual reviews below. Corresponding code and figures are contained in the Supplementary Material. We again want to thank the reviewers for their valuable time, attention and for actively taking part in the review process.

Update:
We have worked in the corresponding changes into our current version of the manuscript and updated the pdf available on openreview.

---

### Meta-Review · Area_Chair_6rHi · 2024-12-17

**Metareview:**

The authors were able to address all issues raised by the reviewers. All reviewers except one were positive about the work. That reviewer was not able to respond, but the authors addressed the issues by them by adding more experiments to evaluate the potential for data contamination, clarify the applicability of assumptions, and discuss alternative data selection strategies.

**Additional Comments On Reviewer Discussion:**

The rebuttal addressed reviewer concerns. The authors re-ran the pipeline with additional random seeds, and implemented alternative item selection strategies. They addressed memorization concerns by testing their benchmark with disjoint sets, permuted answer labels, and fine-tuned models, showing that memorization risks can be mitigated with multiple benchmark versions. The authors also validated their use of Item Response Theory (IRT) and clarified its application to LLM evaluation. In response to suggestions, they conducted simulation studies to explore scenarios involving multiple latent abilities. Overall, the authors provided strong evidence for the reliability and utility of metabench, resulting in an improved consensus towards acceptance.

---

### Decision · Program_Chairs · 2025-01-22

Accept (Poster)